# Barriers and Facilitators of Hepatitis C Care in Persons Coinfected with Human Immunodeficiency Virus

**DOI:** 10.3390/ijerph192215237

**Published:** 2022-11-18

**Authors:** Nir Bar, Noa Bensoussan, Liane Rabinowich, Sharon Levi, Inbal Houri, Dana Ben-Ami Shor, Oren Shibolet, Orna Mor, Ella Weitzman, Dan Turner, Helena Katchman

**Affiliations:** 1Gastroenterology and Hepatology Department, Tel Aviv Medical Center, Tel Aviv 6423906, Israel; 2Sackler Faculty of Medicine, Tel Aviv University, Tel Aviv 6195001, Israel; 3Central Virology Laboratory, Ministry of Health, Chaim Sheba Medical Center, Ramat Gan 5262000, Israel; 4Department of Epidemiology, School of public health, Faculty of Medicine, Tel Aviv University, Tel Aviv 6195001, Israel; 5Center for Liver Disease, Rambam Healthcare Campus, Rappaport Faculty of Medicine, Technion, Haifa 3109601, Israel; 6Crusaid Kobler AIDS Center, Tel Aviv Medical Center, Tel Aviv 6423906, Israel; 7Department of Gastroenterology and Hepatology, Tel Aviv Medical Center, Tel Aviv 6423906, Israel

**Keywords:** low-barrier treatment, hepatitis C infection, human immunodeficiency virus, coinfection, persons who inject drugs, barriers to care

## Abstract

Hepatitis C virus (HCV) and human immunodeficiency virus (HIV) are often co-transmitted. Viral coinfection results in worse outcomes. Persons who inject drugs (PWIDs) face barriers to medical treatment, but HCV treatment is indicated and effective even with ongoing active drug use. We aimed to assess access to HCV care and treatment results in patients coinfected with HIV-HCV. This is a real-world retrospective single-center study of patients followed in the HIV clinic between 2002 and 2018. Linkage to care was defined as achieving care cascade steps: (1) hepatology clinic visit, (2) receiving prescription of anti-HCV treatment, and (3) documentation of sustained virologic response (SVR). Of 1660 patients with HIV, 254 with HIV-HCV coinfection were included. Only 39% of them achieved SVR. The rate limiting step was the engagement into hepatology care. Being a PWID was associated with ~50% reduced odds of achieving study outcomes, active drug use was associated with ~90% reduced odds. Older age was found to facilitate treatment success. Once treated, the rate of SVR was high in all populations. HCV is undertreated in coinfected young PWIDs. Further efforts should be directed to improve access to care in this marginalized population.

## 1. Introduction

Hepatitis C virus (HCV) infection is a significant public health concern that carries a high burden of morbidity and mortality [1,2]. The human immunodeficiency virus (HIV) and HCV viruses share routes of transmission and may result in coinfection. Viral coinfection is common in some populations, and worldwide reports estimate that approximately 25% of persons living with HIV infection are coinfected with HCV [1,3,4].

HCV-HIV coinfection results in worse outcomes compared to HCV mono-infection. Patients with coinfection have increased HCV viremia, more rapid hepatic fibrosis progression, and higher rates of decompensation and death [5]. Successful eradication of HCV infection results in improved survival, reduced morbidity, and improved quality of life [6].

HCV treatment success is defined as sustained virologic response (SVR): a negative HCV polymerase chain reaction (PCR) 12 or 24 weeks after the end of treatment. Until 2013, the main treatment for HCV was based on pegylated interferon (IFN) with ribavirin (RBV) for 24 to 72 weeks. This regimen’s success rates were relatively low (from 40% to 70%, depending on HCV genotype), with high rates of serious adverse events resulting in high drop-out rates [6,7].

The first generation of oral direct-acting antivirals (DAAs), protease inhibitors (telaprevir and boceprevir), was approved in 2011. These medications were combined with IFN and RBV resulting in improved SVR rates but were still associated with many adverse events limiting widespread use.

The next generation of DAAs were available for clinical use in Israel in 2015 and provide a once daily, short duration, oral treatment with a favorable safety profile and outstanding SVR rates (92% to 99%) [8,9]. Multiple clinical trials and real-world cohort analyses showed similar cure rates in HIV-HCV-coinfected patients, compared to HCV mono-infected patients. Near universal SVR rates were found also in those previously considered to have poor prognostic factors such as male sex, certain ethnicity groups, those who were previously treated and in patients with cirrhosis [8].

Unfortunately, HCV treatment initiation rates are still low in patients with HIV, despite regular follow up in HIV clinics and adherence to anti-HIV treatment [10,11].

Persons who inject drugs (PWID) are treated less often than other patients [12]. Medical system misconception, that continued drug use or alcohol consumption will lead to presumed non-adherence, and risk of reinfection may have resulted in exclusion of PWIDs from treatment. In addition, PWIDs are a marginalized population, where individuals may have limited access to basic resources such as food, shelter, safety, and healthcare. This may undermine patients’ perception of self-efficacy to complete an HCV treatment program [10], limiting the probability of achieving the WHO targets of HCV eradication by 2030 [12,13,14,15,16].

In this study, we aimed to explore real world access to HCV care and HCV treatment results in HIV-HCV-coinfected patients, comparing different populations of coinfected patients and focusing on evaluation of social factors that may preclude or facilitate access to care in this population.

## 2. Materials and Methods

### 2.1. Design and Patient Selection

This is a retrospective cohort study of HIV-HCV-coinfected patients in the Tel Aviv Sourasky Medical Center. Patients in the HIV clinic are periodically screened for HCV and other transmittable infections. We included all patients with HIV who underwent HCV screening between 2002 and 2018 and had tested positive for anti-HCV antibody. We examined our records up until December 2020 for HCV-related visits. This time gap was used to allow time for referral to the hepatology clinic, performance of the needed work up, completion of HCV treatment and SVR confirmation in a real-world setting. Epidemiologic, demographic, clinical, and laboratory data were recorded. PWIDs were defined as the exposed group, and the rest of the cohort was used as controls.

During the study period, HCV treatment was centralized and managed in hepatology clinics in Israel. The public health system offered fully subsidized HCV therapies upon recommendation from a hepatology specialist but not an infectious disease specialist. Presently, HCV treatment has been decentralized and is prescribed by the general practitioner in most cases, referring only the rare treatment failures and complex patients to the hepatology specialist.

### 2.2. Definitions

Seroconversion was defined as patients with HIV who had started out as anti-HCV negative but had tested positive for anti-HCV during follow-up. Patients with a detectable viral load by PCR at least 6 months were considered to have chronic HCV infection and require HCV directed viral treatment. The primary outcome was the completion of anti-viral treatment cascade. The stages of the treatment cascade were defined as: (1) having a first hepatology visit (2) completing the work up needed and receiving an anti-viral treatment prescribed (3) completing the treatment course and having a documented SVR at the appropriate time. The cure of HCV, SVR, was defined as a negative HCV PCR 12–24 weeks (depending on regimen) after the end of the anti-viral treatment. We compared linkage to care between PWIDs and controls (i.e., all other patients).

Variables were tested for normality using Kolmogorov–Smirnov testing and Q-Q plots. Categorical variables are reported as number and percentage. Continuous variables are reported as median and interquartile range. T-test or Mann–Whitney test, as needed, were used to compare continuous variables. Fisher exact test and Chi-square test were used to compare the categorical variables. Patients with complete socio-demographic data were used to find predictors (barriers and facilitators) for achieving the primary outcome. Univariate logistic regression was first performed. Age, sex, and other predetermined potential barriers and facilitators (age, sex, family status, PWID, active drug use, and mode of transmission) were examined in a multiple variables forward stepwise logistic regression model to further examine the importance of these predictors. All statistical tests were 2-sided, and *p*-value < 0.05 was considered statistically significant. SPSS software was used for all statistical analysis (IBM SPSS statistic, version 22, 2013, IBM Corp, Armonk, NY, USA).

## 3. Results

### 3.1. Baseline Characteristics

Between the years 2002 and 2018, a total of 1660 HIV patients were followed and screened for HCV in the HIV clinic. We found 274 (16.5%) patients with positive anti-HCV during routine screening. The vast majority (241/274, 87.9%) of the patients were diagnosed simultaneously with HIV and HCV. The additional 33 (12%) patients demonstrated HCV seroconversion (new acquisition of HCV during follow up). In 16 (6%) of the 274 anti-HCV positive patients, the HCV PCR was negative at the time of diagnosis (first PCR), and in an additional 4 (1.5%), there was a documented spontaneous recovery (initial positive PCR which turned negative with no HCV directed treatment).

Concurrent chronic HBV infection (positive HBV surface antigen (HBsAg)) was documented in 14 (5.1%) out of 274 coinfected patients and evidence of resolved HBV infection (positive anti-HBV core (HBc) and negative HBsAg) in 156 (56.9%) patients. All the patients received anti-retroviral therapy with undetectable HIV viral load and median (IQR) CD4 count 594 (415–818).

Baseline characteristics of the 274 patients are summarized in Table 1. Most of the patients were male immigrants, predominantly from the former Soviet Union. The most common presumed route of HIV and HCV transmission was a history of substance use or sexual transmission. Of the 135 PWIDs, 39 (14.2%) patients were active substance users at the time of the follow up.

### 3.2. HCV Treatment Protocols and Results

Forty-three patients received their treatment in the pre-DAA era (PEG + RBV) with SVR achieved in eighteen (41.8%). Eight additional patients were treated with 1st generation DAA (two were treatment naïve, and six failed on PEG + RBV therapy). Seven achieved documented SVR, and one had viral breakthrough during the treatment.

Eighty-six patients were treated with DAA (74 treatment naïve and 12 PEG/RBV or 1st generation DAA failure). In 72 patients that underwent HCV PCR 12 weeks after the end of treatment, 100% SVR (72/72) was documented. An additional 14 patients were lost to follow up and no HCV PCR was performed, bringing intention to treat SVR to 84% (72/86). Two patients that achieved documented SVR were reinfected with HCV and received an additional treatment course with DAA. One of them had documented SVR, and the other lost to follow up.

### 3.3. Linkage to Hepatitis C Care

We analyzed the access to care cascade in all the patients with positive HCV PCR (n = 254). Although regular follow up in the HIV clinic was documented in 214 (84%) of coinfected patients, only 127 (50%) visited the hepatology clinic at least once. Of the patients that attended the hepatology clinic, 111/127 (87.4%) completed the work up for the treatment and received a prescription for anti-HCV medications. Of the patients who received a prescription, 98/111 (88%) started the treatment and had documented SVR. Therefore, only 98/254 (38.5%) of the patients with HIV HCV coinfection with a positive HCV PCR successfully completed the HCV care cascade.

Complete demographic data were available in 184 patients, included in the barriers and facilitators to HCV care analysis. We compared patients according to the route of HIV transmission (PWIDs vs. controls). The results are presented in Table 2. The control consisted mostly of patients who contracted HIV through unprotected sex: 26 (53.1%) were men who have sex with men (MSM) and 14 (28.6%) non-MSM unprotected sex. Of these 14 patients, 1 (2%) was a sex worker. Some PWID had overlapping risk factors: three (2.2%) were sex workers, and one (0.7%) also reported non-MSM unprotected sex.

Age and sex were similar in both groups as well as marital status and having children. However, the proportion of immigrants was higher in PWID than in the non-PWID group. The genotype distribution differed between the groups (see Table 2). Fibrosis level was similar, as were the rates of triple infection. Exposure to syphilis during follow-up was higher in the non-PWID group.

In this cohort, first hepatology clinic visit, receiving a prescription, and documented SVR were achieved in 103 (76.3%), 101 (74.8%), and 80 (59.3%), respectively. When comparing PWID vs. non-PWID, the outcomes were achieved in 67 (49.6) vs. 36 (73.5%), 65 (48.1%) vs. 36 (73.5%), and 50 (37%) vs. 30 (61.2%), respectively *p* < 0.01 for all comparisons. Figure 1 represents the comparison of access to care cascade in the PWID vs. non-PWID population.

We used logistic regression analysis to find predictors for having completed the cascade of care steps (visit in the hepatology clinic, getting anti-HCV treatment, achieving documented SVR). The univariate analysis was performed in 184 patients with complete socio-demographic data about route of viral transmission, see Table 3. Being a PWID was found to be a major barrier to access to care (i.e., compared to control, PWIDs had lower odds of achieving every study outcome). Thus, for the PWID group, the OR (95% CI) for completing a first clinic visit, receiving anti-HCV prescription, and a documented SVR was: 0.36 (0.17–0.73), 0.34 (0.16–0.69), and 0.37 (0.19–0.73), *p* < 0.01 for all. Active drug use was an additional barrier to linkage to care with an OR (95% CI) of active users for clinic visit, receiving anti HCV prescription, and a recorded SVR of 0.1 (0.04–0.26), 0.08 (0.03–0.23), 0.08 (0.02–0.29), *p* < 0.001 for all, respectively.

When comparing active and former drug users, we found a striking difference in achieving the primary access to care (first clinic visit): 6 (15%) vs. 61 (64.2%), *p* < 0.001, respectively. Furthermore, getting anti-HCV treatment, having a documented SVR was achieved in 5 (12.5%) vs. 60 (63.2%) and 3 (7.5%) vs. 47 (49.5%), *p*< 0.001 for both, respectively.

Older age was the only facilitator (i.e., associated with significant outcomes improvement). Interestingly, sex, marital status, smoking status, or drinking habits had no significant association with access to care outcomes.

In the multivariate analysis, age, being PWID as a presumed route of HCV acquisition, and active drug use were associated with decreased access to HCV care (Table 4).

Of note, once having started the treatment and remaining in follow up, the rates of SVR were similar in both populations (100% SVR for both groups with DAA treatment). The rate of reinfection after SVR was also similar (one patient in each group).

## 4. Discussion

In the era of highly effective anti-HCV therapy, high SVR rates have become standard of practice both in mono-infected HCV patients and in patients coinfected with HCV and HIV [17] and access to care has become the rate-limiting factor to successful anti-viral treatment [18].

In this study, we report results of anti-HCV treatment of HIV-HCV-coinfected patients in the Tel Aviv Sourasky Medical Center HIV clinic and examined the major barriers and facilitators related to treatment success. The Israeli health care system offers universal health insurance and individual copay costs are quite low, even for DAA therapies. Despite this, we found the uptake of HCV care in this population to be low, with documented HCV cure achieved in only 39% (99 out of 254) of the screened patients with positive HCV PCR. However, once coinfected patients were linked to hepatology care and remained in follow up, there was a 100% SVR rate with all DAAs.

Our data suggest that the main gap in the access to care cascade in coinfected PWIDs is the need for a specialist visit; the first hepatology visit took place only in 50% of coinfected patients (127/254). Once the linkage to specialist care was done, the rate of achieving the next step in care cascade (receiving a prescription) was close to 100%. These findings are concordant with previous studies, reporting an HCV prescription rate between 43% and 60% in HIV-HCV-coinfected patients [19,20,21] and a rate of SVR that is identical to mono-infected population [22]. This significant gap in linkage to specialist care, described in our and previous studies, emphasizes the importance of decentralization of HCV care, bringing HCV treatment to HIV clinical providers, general practitioners, or addiction care centers and was shown to improve treatment rates [23,24].

When examining the main barriers to HCV care access in our population, history of injection drug use was found to be an independent barrier to care, with the odds of achieving the cascade outcomes 50–60% lower compared to controls. Active drug use was an even greater barrier with ~90% reduced odds of achieving these outcomes.

Our findings confirm previous reports showing IV drug use is associated with reduced HCV treatment uptake [25]. Saab et al. found that almost half of patients not referred to an HCV specialist were PWIDs, compared to 17% of those who were referred. The decreased likelihood of anti-HCV medication prescriptions among patients with a history of substance use might reflect provider reluctance because of concern about non-adherence, stigma against substance use, and concern about the risk of reinfection [26]. Importantly, our study demonstrated only two cases of reinfection, keeping with other reports of very low rates of HCV reinfections even in the setting of active injection drug use [27].

Our patient population had a high proportion of immigrants from the former Soviet Union. This is consistent with established risk factors for HCV positivity in the general population in Israel which include a history of blood products transfusion before 1990, history of drug use, and being an immigrant from the former Soviet Union [28]. Immigrants have additional unique barriers that may contribute to the difficulty in access to care: including language-related barriers, cultural differences, discrimination, and racism [29]. In our population, there was no significant difference between Israeli-born or immigrants in access to care

Our finding of younger coinfected individuals being less likely to have received HCV treatment is consistent with the results of previous studies. Corcorran et al. demonstrated a 5% increase in the odds of having been prescribed a DAA for every year age increase [30]. Moris et al. have shown very low intake of treatment in patients under age 30 [31]. These findings are very concerning given the rapid increase in acute HCV among PWID less than 40 years of age [32,33] that can maintain the reservoir of HCV transmission.

New strategies to improve younger PWIDs linkage to care should be found, for example engaging PWIDs to educate peers about the new treatment modalities and share their treatment success, which may address distrust some individuals may have with an ostracizing medical system [34,35].

Our study has several limitations. The inherent limitations of a retrospective study design could include missing data and unaccounted confounders affecting outcome estimation. However, as with all studies focusing on populations that are often excluded from prospective studies (even if not officially), our data reflect real life experience. Our focus on patients who are followed in our HIV clinic with complete data was done to address the patients who may have been referred for treatment in other centers.

Our study confirms the challenges marginalized populations are still facing in achieving access to anti-HCV treatment. Significant social barriers should be overcome in these subgroups, who can potentially achieve high SVR rates. Physician education and HCV treatment decentralization are crucial in achieving these goals.

## 5. Conclusions

We have found that PWIDs, especially those with active drug use, face barriers in linkage to care and are undertreated for a curable disease. Our study confirms the challenges marginalized populations are still facing in achieving access to anti-HCV treatment. Significant social barriers should be overcome in these subgroups, who can potentially achieve high SVR rates. Physician education and HCV treatment decentralization are crucial in achieving these goals.

## Figures and Tables

**Figure 1 ijerph-19-15237-f001:**
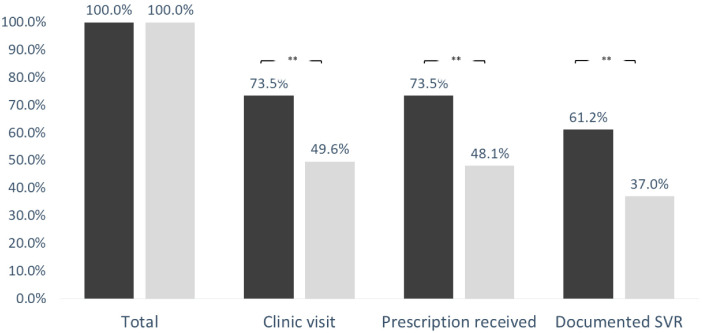
Access to hepatitis C care cascade. Study outcomes achieved by persons who inject drugs compared to other controls. PWID—persons who inject drugs; **—*p* < 0.01.

**Table 1 ijerph-19-15237-t001:** Basic characteristics of the study population.

	Total (n = 274)
Age (years)	37.2 ± 8.6
Male sex	202 (73.7)
Immigrant from endemic country	219 (79)
Excessive Alcohol use	35 (12.7)
Active substance use	40 (14.6)
Opiate substitution treatment	24 (9)
Presumed mode of transmission (n = 184), some coexist.	
PWID	135 (73.4)
MSM	28 (15.2)
Unprotected intercourse (non MSM)	18 (9.7)
Blood product transmission	8 (4.3)
Iatrogenic transmission	1 (0.5)
HCV characteristics (n = 142)	
1a	20 (14.1)
1b	64 (45.1)
2	5 (3.5)
3	38 (26.7)
4	15 (10.6)
Liver fibrosis level (n = 109)	
F0-1	66 (60.5)
F2	16 (14.7)
F3	11 (10.1)
F4	16 (14.7)
Triple infection (HBsAg positive)	14 (5.1)
Resolved HBV infection (HBsAg negative, HB-core positive)	156 (56.9%)

Values are presented in mean ±SD or median (IQR), n (%) as needed. PWID—persons who inject drugs; MSM—men who have sex with men; HCV—hepatitis C virus; HBsAg—hepatitis B surface antigen.

**Table 2 ijerph-19-15237-t002:** Comparison of the basic characteristics of two study populations.

	Non-PWID (n = 49)	PWID (n = 135)	*p* Value
Age (years)	37.8 + 8.2	36.1 + 13.3	NS
Age at HIV diagnosis	33.8 ± (8.6)	33.9 ± (8.8)	NS
Male sex	37 (75.5)	101 (74.8)	NS
Marital status			NS
Single	26 (53.1)	63 (46.7)	
Married	11 (22.4)	43 (31.9)	
Divorced/widowed	12 (24.5)	29 (21.5)	
Has children	20 (40.8)	73 (54.1)	NS
Active drug use	Not applicable	40 (29.6)	NA
Excessive Alcohol use	8 (28.6)	27 (38.6)	NS
Immigrant from endemic country	29 (59.2)	119 (88.1)	*p* < 0.001
Presumed viral transmission way			NA
MSM	26 (53.1)	2 (1.5)	
Non MSM unprotected sex (see text)	14 (28.6)	4 (3.0)	
Blood product transmission	8 (16.3)	0 (0)	
Iatrogenic transmission	1 (2)	0 (0)	
HCV genotype (n = 118)			<0.001
1a	9 (23.7)	10 (12.5)	
1b	14 (36.8)	37 (46.3)	
2	0 (0)	4 (5)	
3	4 (10.5)	27 (33.8)	
4	11 (28.9)	2 (2.5)	
CD4 count	620 (365–818)	594 (436–831)	
Fibrosis level (n = 109)			NS
F0-1	22 (61.1)	44 (60.3)	
F2	5 (13.9)	11 (15.1)	
F3	4 (11.1)	7 (9.6)	
F4	5 (13.9)	11 (15.1)	
Serologic data			
HBs Ag (+): triple infection (data available in 174)	3 (6.8)	7 (5.4)	NS
Anti-HB core (+) (data available in 173)	25 (56.8)	92 (71.3)	NS
VDRL (data available in = 154)	10 (25.6)	9 (7.8)	0.001
TPHA (data available in = 154)	16 (41)	18 (15.7)	0.009

Values are presented in mean ±SD or median (IQR), n (%) as needed. PWID—persons who inject drugs; HIV—human immunodeficiency virus; MSM—men who have unprotected sex with men; unprotected sex—see text; HCV—hepatitis C virus; HBsAg—hepatitis b surface antigen; Anti-HB core—hepatitis B core antibody; VDRL—venereal disease research laboratory test; TPHA—treponema pallidum hemagglutination test; NS—non-significant; NA—non applicable.

**Table 3 ijerph-19-15237-t003:** Univariate analysis.

	Clinic Visit	Prescription Received	SVR Documentation
Age (for every 5y)	1.17 (1.01–1.34), *p* = 0.031	1.16 (1.01–1.33), *p* = 0.036	1.18 (1.02–1.36), *p* = 0.025
Male sex	1.36 (0.79–2.33), NS	1.3 (0.76–2.24), NS	1.45 (0.81–2.57), NS
Being married	1.03 (0.71–1.48), *p* NS	1.08 (0.75–1.56). NS	1.18 (0.82–1.7), NS
Having children	1 (0.56–1.78), NS	1.09 (0.61–1.94), NS	1.26 (0.7–2.3), NS
Being an Immigrant from endemic countries	0.54 (0.3–1), *p* = 0.051	0.58 (0.32–1.06), *p* = 0.077	0.63 (0.34–1.16), *p* = 0.135
Excessive Alcohol use	0.81 (0.37–1.74), NS	0.81 (0.37–1.74), NS	1.11 (0.52–2.39), NS
Being a PWID	0.36 (0.17–0.73), *p* = 0.005	0.34 (0.16–0.69), *p* = 0.003	0.37 (0.19–0.73), *p* = 0.004
Active drug use	0.1 (0.04–0.26), *p* < 0.001	0.08 (0.03–0.23), *p* < 0.001	0.08 (0.02–0.29), *p* < 0.001

This table combines the univariate logistic regression for all linkage to care outcomes. Each outcome is presented in a different column. For each outcome, the OR (95% confidence interval) *p* values are provided. SVR—sustained virologic response; PWID—persons who inject drugs; bold boxes denote significance.

**Table 4 ijerph-19-15237-t004:** Multivariate analysis.

	Clinic Visit	Prescription Received	SVR Documentation
Age (for every 5y)	1.36 (1.11–1.68) *p* = 0.003	1.35 (1.1–1.66) *p* = 0.004	1.25 (1.04–1.52) *p* = 0.021
PWID	0.44 (0.2–0.98), *p* = 0.043	0.41 (0.18–0.91) *p* = 0.028	0.45 (0.22–0.94) *p* = 0.034
Active drug use	0.1 (0.42–0.24), *p* < 0.001	0.09 (0.04–0.22) *p* < 0.001	0.1 (0.04–0.27) *p* < 0.001

This table presents the multiple variable logistic regression for all linkage to care outcomes. Each outcome is presented in a different column. For each outcome, the adjusted OR (95% confidence interval), *p* value are shown after adjusting for age, sex, family status, PWID, active drug use, and route of transmission. Only significant associations are provided. PWID—patient who injects drugs.

## Data Availability

Since HIV details are confidential in Israel, data are not publicly archived. For access to partial deidentified data, contact the corresponding author.

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
