# Peer review of "Barriers and Facilitators of Hepatitis C Care in Persons Coinfected with Human Immunodeficiency Virus"

_ijerph, 2022, doi:10.3390/ijerph192215237_

Round 1
Reviewer 1 Report
This is an important real-world analysis of the HCV care cascade in persons coinfected with HIV and HCV. My main concern is that in many places in the world, the physicians who are providing HIV care also treat hepatitis C. The call to decentralize HCV treatment care makes no sense from this perspective because it is clear that this center is doing an excellent job with HIV care and the obvious question for readers will be why they don’t also do the (now very easy) HCV care. If there is a policy in Israel that infectious disease and HIV specialists are not allowed to provide HCV care, that needs to be explained, that policy obviously needs to be questioned since it is clearly harming people, and it begs the question of how “decentralized” other providers would be allowed to provide HCV treatment.
Minor comments:
Page 1: The variation in capitalization is quite distracting and annoying (i.e. “care” in the title, “university” in affiliations, The Human immunodeficiency virus, etc). Please go through this document, including all affiliations, and correct capitalization.
Abstract: The abstract provides dates of 2002 and 2017, but in the body of the manuscript I see 2002 to 2018 and a mention of 2020. Please reconcile dates.
Abstract: “facilitator treatment success” is defined in the body of the manuscript (page 6) but here it is not obvious what this means. Can you change to “facilitating treatment success”?
Page 1: “share similar ways of transmission…” Ways sounds odd. Consider routes of transmission (also see page 4).
Page 2: Were next gen DAAs approved in Israel in 2015 (there were approved in US in 2014)? If so please specify location of approval. Sofosbuvir/velpatasvir is associated with SVR rates of 99%, not 97%.
Table 1: It seems odd that the HIV characteristics (CD4 and viral load) of this HIV-HCV cohort were in the text and not the table. Not a major concern. Is it true that 100% of this entire cohort of people with HIV (including the PWID) had undetectable HIV RNA levels? That demonstrates outstanding care.
Author Response
Reviewer 1
This is an important real-world analysis of the HCV care cascade in persons coinfected with HIV and HCV. My main concern is that in many places in the world, the physicians who are providing HIV care also treat hepatitis C. The call to decentralize HCV treatment care makes no sense from this perspective because it is clear that this center is doing an excellent job with HIV care and the obvious question for readers will be why they don’t also do the (now very easy) HCV care. If there is a policy in Israel that infectious disease and HIV specialists are not allowed to provide HCV care, that needs to be explained, that policy obviously needs to be questioned since it is clearly harming people, and it begs the question of how “decentralized” other providers would be allowed to provide HCV treatment.
Reply: We wholeheartedly agree with this remark about the need of decentralizing the therapy of HCV if we are to achieve the goal of HCV eradication. In Israel the public health system required a hepatologist to prescribe HCV therapy in the first years when therapy became available. We added this explanation to the manuscript. Thank you for this important comment. See page 2: “…During the study period, HCV treatment was centralized and managed in hepatology clinics in Israel. The public health system offered fully subsidized HCV therapies upon recommendation from a hepatology specialist but not an infectious disease specialist. Presently, HCV treatment has been decentralized and is given by the general practitioner in most cases, referring only the rare treatment failures and complex patients to the hepatology specialist.”
Minor comments:
Page 1: The variation in capitalization is quite distracting and annoying (i.e. “care” in the title, “university” in affiliations, The Human immunodeficiency virus, etc). Please go through this document, including all affiliations, and correct capitalization.
Reply: We thank the reviewer for this observation. We have revised the manuscript accordingly.
Abstract: The abstract provides dates of 2002 and 2017, but in the body of the manuscript I see 2002 to 2018 and a mention of 2020. Please reconcile dates.
Reply: We thank the reviewer for noticing this ambiguity. We had meant 2018 and revised the abstract. In addition, we revised the methods section to clarify that the gap between 2018 and 2020 was chosen to allow patients to complete the work up needed before treatment. See page 2, under 2.1. design and patient selection: “… This time gap was used to allow time for referral to the hepatology clinic, performance of the needed work up, completion of HCV treatment and SVR confirmation…”
Abstract: “facilitator treatment success” is defined in the body of the manuscript (page 6) but here it is not obvious what this means. Can you change to “facilitating treatment success”?
Reply: Thank you for this suggestion, we revised the abstract: “…Older age was found to facilitate treatment success.”
Page 1: “share similar ways of transmission…” Ways sounds odd. Consider routes of transmission (also see page 4).
Reply: Thank you for this comment, we revised the manuscript and changed all “ways of transmission” to routes of transmission.
Page 2: Were next gen DAAs approved in Israel in 2015 (there were approved in US in 2014)? If so please specify location of approval. Sofosbuvir/velpatasvir is associated with SVR rates of 99%, not 97%.
Reply: Thank you for the comment, next generation DAA become available in Israel in 2015 and it's clarified now in line 55. The second point is clarified in line 57: “…The next generation of DAAs were available for clinical use in Israel in 2015 and provide a once daily, short duration, oral treatment with a favorable safety profile and outstanding SVR rates (92% to 99%).”.
Table 1: It seems odd that the HIV characteristics (CD4 and viral load) of this HIV-HCV cohort were in the text and not the table. Not a major concern. Is it true that 100% of this entire cohort of people with HIV (including the PWID) had undetectable HIV RNA levels? That demonstrates outstanding care.
Reply: Thank you. This is true but please note the cohort includes those who are in regular follow up and underwent serial anti-HCV screening. Other patients exist who are lost to follow-up, or did not comply with the screening protocol who could have detectable HIV viral loads
Reviewer 2 Report
The work is interesting and generally I have no critical remarks, although the results are quite obvious and easy to predict.
Additionally, I believe that one of the conclusions should be that, the drug addiction treatment prior to qualification for HCV therapy should be provided.
Perhaps it would also be worth comparing SVR in active and former drug users, and not only active drug addicts with non-PWID.
Author Response
Reviewer 2
The work is interesting and generally I have no critical remarks, although the results are quite obvious and easy to predict.
Additionally, I believe that one of the conclusions should be that, the drug addiction treatment prior to qualification for HCV therapy should be provided.
Perhaps it would also be worth comparing SVR in active and former drug users, and not only active drug addicts with non-PWID.
Reply: Thank you. We added the comparison of active vs. former drug users to the results. See page 6 lines 198-202: “…When comparing active and former drug users, we found a striking difference in achieving the primary access to care (first clinic visit): 6 (15%) vs 61 (64.2%), p<0.001, respectively. Furthermore, getting anti-HCV treatment, having a documented SVR was achieved in 5 (12.5%) vs 60 (63.2%) and 3 (7.5%) vs. 47 (49.5%), p< 0.001 for both, respectively.”
Reviewer 3 Report
The present retrospective cohort study present data about the access to HCV care and treatment results in patients coinfected with HIV/HCV , in Tel Aviv Sourasky Medical Center, between the years 2002-2018, a total of 1660 HIV patients were followed and screened 112 for HCV in the HIV clinic.
The data presented are relevant, and could support the clinical decision in others coinfected person around the world.
Suggestions:
- Is necessary an extensive revision of English language.
- The table 3 and 4 are very unclear, the authors could improve the tabel design.
Author Response
Reviewer 3
The present retrospective cohort study present data about the access to HCV care and treatment results in patients coinfected with HIV/HCV , in Tel Aviv Sourasky Medical Center, between the years 2002-2018, a total of 1660 HIV patients were followed and screened 112 for HCV in the HIV clinic.
The data presented are relevant, and could support the clinical decision in others coinfected person around the world.
Suggestions:
- Is necessary an extensive revision of English language.
- The table 3 and 4 are very unclear, the authors could improve the tabel design.
Reply: Thank you for this comment, the manuscript was revised by native English speaker colleague. In addition, we amended the legends for table 3 and table 4 (see text).